# T2LM: Long-Term 3D Human Motion Generation from Multiple Sentences

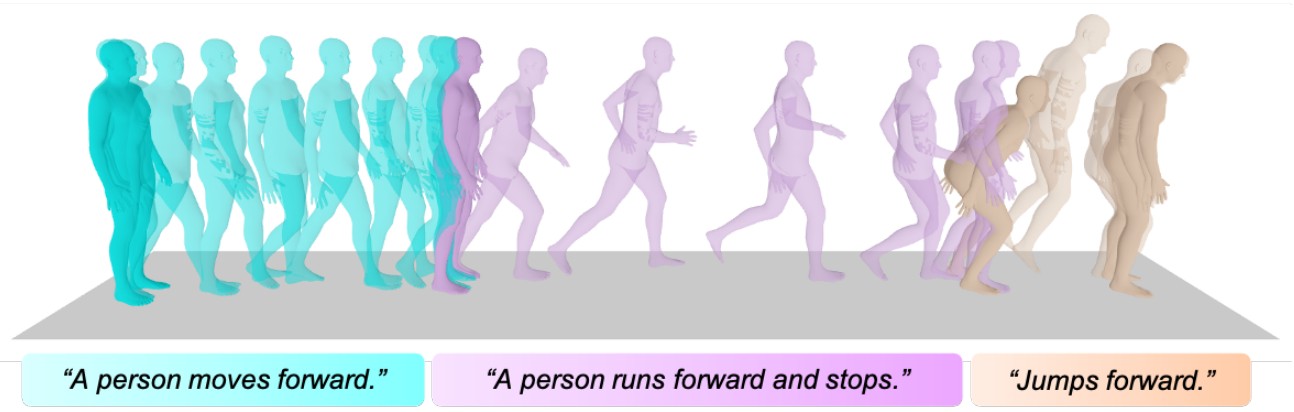

*"A person moves forward."*     *"A person runs forward and stops."*     *"Jumps forward."*

Figure 1. **Visual result.** We present a qualitative example obtained from our long-term motion generator. A stream of input texts is used to condition our model and produce a matching continuous motion.

## Abstract

*In this paper, we address the challenging problem of long-term 3D human motion generation. Specifically, we aim to generate a long sequence of smoothly connected actions from a stream of multiple sentences (i.e., paragraph). Previous long-term motion generating approaches were mostly based on recurrent methods, using previously generated motion chunks as input for the next step. However, this approach has two drawbacks: 1) it relies on sequential datasets, which are expensive; 2) these methods yield unrealistic gaps between motions generated at each step. To address these issues, we introduce simple yet effective **T2LM**, a continuous long-term generation framework that can be trained without sequential data. **T2LM** comprises two components: a 1D-convolutional VQVAE, trained to compress motion to sequences of latent vectors, and a Transformer-based Text Encoder that predicts a latent sequence given an input text. At inference, a sequence of sentences is translated into a continuous stream of latent vectors. This is then decoded into a motion by the VQVAE decoder; the use of 1D convolutions with a local temporal receptive field avoids temporal inconsistencies between training and generated sequences. This simple constraint on the VQ-VAE allows it to be trained with short sequences only and produces smoother transitions. **T2LM** outperforms prior long-term generation models while overcoming the constraint of requiring sequential data; it is also competitive with SOTA single-action generation models.*

## 1. Introduction

Human motion generation plays a vital role in numerous applications of computer vision [9, 20, 54] and robotics [11, 28, 44, 47]. Recent trends focus on controlling generated human motions with input prompts such as discrete action labels [14, 31, 32, 37, 56], or free-form text [15, 16, 38, 40, 49, 60, 61]. However, controllable synthesis of *long-term* human motion is less studied [5, 46] and remains challenging, mainly due to the scarcity of long-term training data. In this work, we propose a model to produce *long-term human motion* from a given stream of textual descriptions of *arbitrary* length without requiring sequential data for training.

Real-life human motion is continuous and can be viewed as a temporal composition of *actions*, with *transition* in between. Although the text-conditional generation of short *actions* has been thoroughly addressed by previous work [37, 38, 51], modeling smooth and realistic *transitions* remains a core challenge for generating long-term motions

| Method | Trained without sequential data | Continuous generation |
|---|---|---|
| TEACH [5] | ✗ | ✗ |
| MultiAct [22] | ✗ | ✗ |
| ST2M [25] | ✗ | ✗ |
| DoubleTake [46] | ✓ | ✗ |
| **T2LM (Ours)** | ✓ | ✓ |

Table 1. **Comparison to previous methods.** T2LM can be trained without sequential datasets such as BABEL. Previous models with discontinuous decoding generate unrealistic gaps between the consecutive actions. In contrast, our approach employs a continuous decoding scheme for smoother transitions between actions.

usable in practical applications [33].

While a body of work [5, 22, 25, 46] on long-term motion generation has been introduced, we identify two limitations of these methods summarized in Table 1. First, existing methods such as MultiAct [22], TEACH [5], or ST2M [25] rely on sequential data for training. Compared to single-action datasets [14, 15], which contain annotations for short actions, a sequential dataset [41] contains frame-level annotations for each individual action and transition within long-term motion. While this provides valuable data to capture how *transitions* connect consecutive *actions*, acquiring such dense frame-level annotation at scale is expensive, and determining the segment between actions is not trivial. In addition, capturing transitions for all possible pairs of actions at scale is impossible. This dependency limits the applicability of existing methods to new domains.

Second, existing methods empirically struggle to create smooth and realistic transitions. We hypothesize this is due to discontinuities in the generation process when chaining actions together. The majority of works [5, 22, 25] recurrently generates the long-term motions at two granularities: actions of each step are conditioned on the output of the previous step, and those actions are concatenated into long-term motion. Concurrently, DoubleTake [46] uses the MDM [51] to generate actions independently and blends them into a long-term motion with a diffusion model. This approach also operates at two granularities, generating individual actions and merging them. It results in abrupt speed changes and discontinuities between consecutive actions. In this work, we hypothesize that a framework that instead stays at a single granularity can alleviate these issues and generate smoother transitions.

As illustrated in Fig. 2, we propose a conceptually simple yet effective framework **T2LM**. Our method a) can generate a motion continuously across the input sentences and b) does not require long-term action sequences for training, thus overcoming the limitations of existing work. At train time, we first train VQVAE to map an input motion into a sequence in a discrete latent space. The mapped latent sequence is used as a target for a *Text Encoder*, a text-and-

length conditional latent prediction model. Both are trained with single actions and accompanying texts. At inference time, a stream of input sentences and desired motion lengths is encoded into a stream of latent vectors. Finally, we continuously reconstruct the desired long-term motion with the 1D convolutional decoder.

Our model has two key properties: First, it produces sequences of latent vectors, unlike approaches that encode the entire sequence into a single latent vector like Actor [37]. Second, we learn a prior over small chunks of motion, each encoded independently from the others, using a VQVAE encoder built from 1D convolutional layers with a local receptive field. This assumption, which departs from methods taking all past motion into account like PoseGPT [31], is the simplest way to avoid any discrepancies between short training sequences and long sequences at inference time.

These two key properties offer several advantages for long-term generations. First, it is possible to process a sequence of infinite length on the fly, as the cost of forwarding the model is linear in the size of the local receptive field [42]. This is in contrast with methods that employ a vanilla transformer architecture with a complexity that is quadratic in the sequence length. Thus, our model can process a continuous stream rather than a sequence of chunks that have to be later post-processed [5]. Secondly, using a sequence of latents with local receptive field allows to convey fine-grained semantics at the right temporal location. Empirically, we show that these simple changes lead to higher-quality actions compared to existing methods that generate variable-length actions with a single latent vector.

Our experiments show that **T2LM** outperforms the state-of-the-art on long-term generation while matching or outperforming existing approaches for single-action when evaluated with FID scores and R-precision. We present two novel metrics aimed at evaluating the quantitative excellence of long-term motion more effectively: a) during transitions and b) along the sequence utilizing a sliding window approach.

Our contributions are the following:
- We propose a conceptually simple yet effective method **T2LM** for generating long-term human motions from a continuous stream of arbitrary-length text sequences.
- We make two architectural design choices which together enable **T2LM** to generate smooth transitions and to be trained without any long-term sequential training data.
- As a result, **T2LM** outperforms previous long-term generation methods while overcoming their limitations. We also match the performance of previous state-of-the-art single-action generation models.

## 2. Related works

**Human motion synthesis.** Human motion synthesis is naturally formulated as a generative modeling problem. In

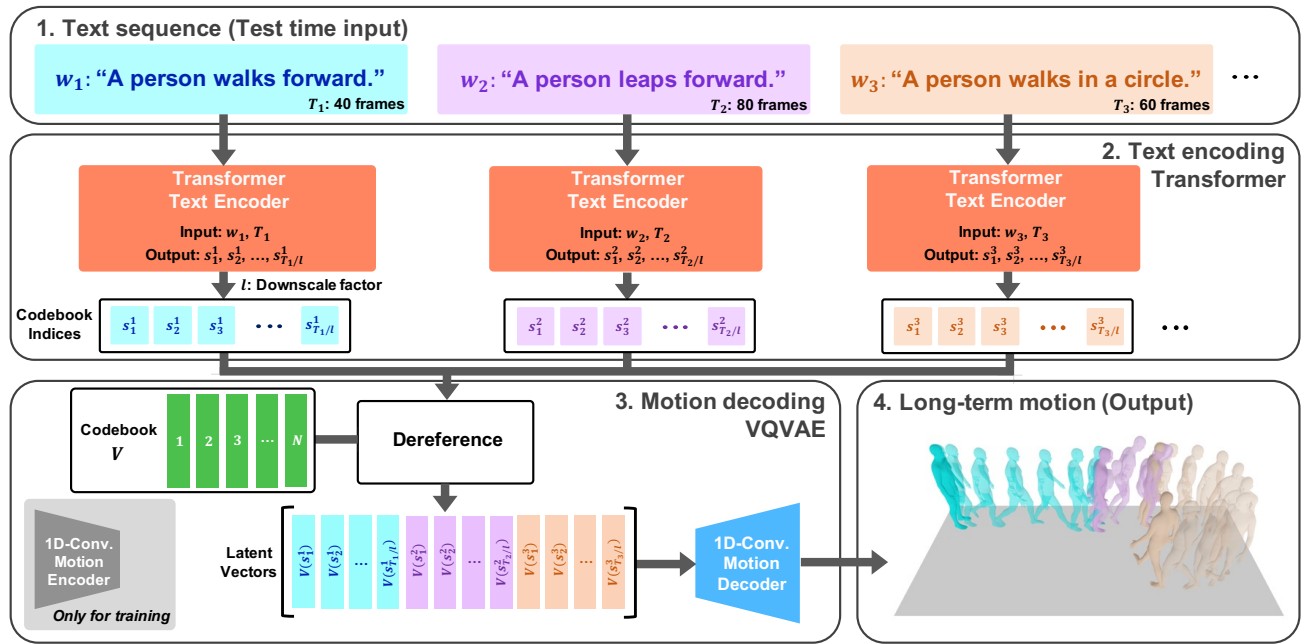

Figure 2. **Overview of T2LM.** We present the overview of our test-time generation. From the stream of textual descriptions and desired lengths of each action, we produce a smooth long-term motion corresponding to the text stream.

particular, prior works have relied on Generative Adversarial Networks (GANs) [1, 27], Variational Auto-encoders (VAEs) [14, 37], Normalizing flows [19, 59], diffusion models [46, 51, 52, 58], or the VQ-VAE framework [22, 31, 60, 63]. Motion can be predicted from scratch or given observed frames, from the past only [4, 17, 36, 57, 62], or also with future targets [10, 18]. Other forms of conditioning can be used, such as speech [7, 13], music [21, 23, 24], action labels [14, 31, 37], or text [1, 3, 12, 26, 27, 45]. In the presence of text inputs, human motion generation can also be cast into a machine-translation problem [1, 26, 39]; a joint cross-modal latent space can also be used [3, 12, 55]. In this work, we consider motion generation conditioned on text sentences from a generative modeling perspective.

**Action and text conditioned human motion generation.** Early action conditional motion models relied on Conditional GANs [8] and conditional VAEs [14, 32, 37]. More flexible variants have been proposed using the VQ-VAE framework; in particular, PoseGPT [31] allows conditioning on past observations relying on a GPT-like model to sample motions. Human motion can be generated conditionally on text. Earlier works include the Text2Action model [2], based on an RNN conditioned on a short text. Motion-CLIP [50] aligns text and motion by leveraging the powerful CLIP [43] model as the text encoder and empirically shows that this enables out-of-distribution motion generation. TEMOS [38] extends the VAE-based approach AC-TOR [37] to obtain a text-conditional model using an additional text encoder. T2M [15] proposed a large-scale dataset called HumanML3D, which is better suited to the task of text-conditional long motion generation. TM2T [16] jointly considers text-to-motion and motion-to-text predictions and shows performance gains from jointly training both tasks. Recently, T2M-GPT [60] have achieved competitive performance using the VQ-VAE framework, where motion is encoded into discrete indices, which are then predicted using a GPT-like model. Diffusion-based models have also emerged as a powerful class of models to generate motion conditionally on text [51]. Related to our works, Multi-Act [22], ST2M [25] and TEACH [5] utilize a recurrent generation framework with past-conditional VAE to generate multiple actions sequentially. These require sequential training data [41], an inherent limitation of the recurrent paradigm. DoubleTake, a part of PriorMDM [46] that utilizes MDM [51] as a generative prior, individually generates the actions and connects them with a diffusion model.

# 3. Method

We now present in detail our **T2LM** approach. First, we explain how we compress human motion into a discrete space and reconstruct motion from it (Sec. 3.1). Second, we introduce a GPT-like autoregressive Text Encoder designed to map a given text to a sequence in the discrete latent space learned by the VQ-VAE (Sec. 3.2). Third, we discuss in Sec. 3.3 our procedure to generate long-term motion sequences corresponding to input text streams. We also include a desired length for each action in the stream. At train

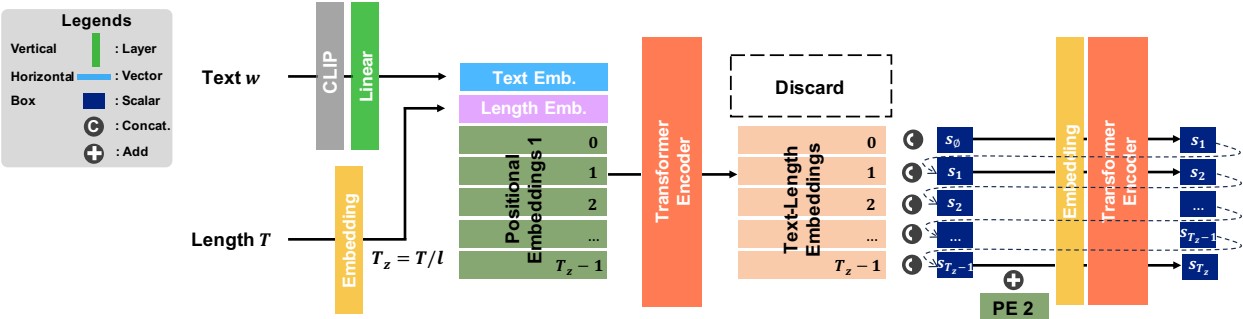

Figure 3. **VQVAE architecture.** We present the architecture of our VQVAE. Both the encoder and the decoder are built with convolutional layers.

Figure 4. **Text Encoder architecture.** We present the architecture of Text Encoder. A first test encoder injects information about the text and length embeddings into a sequence of tokens, and a second autoregressive model predicts the latent sequence.

time, this is extracted from the data, while at inference, this can be either treated as an input or sampled from a prior.

### 3.1. Learning a discrete latent representation

**Motivation.** Human motion is typically represented as a temporal sequence of 3D points – human meshes or skeletons – or a sequence of model parameters that produce such 3D representations [29, 35]. Plausible human motion usually represents a very small portion of these representation spaces, as evidenced by the fact that sequences of random samples do not produce any realistic motion. This has motivated methods that compress human motion into a discrete latent space and has shown to be beneficial for reconstruction and manipulation [31, 60]. In contrast to previous approaches [5, 38, 46, 51], where a single latent represents the entire action available at each step, we design our approach so that each latent represents a fixed length of human motion. This enables continuous decoding of the semantics from textual descriptions without creating a duration mismatch between train and test sequences. We employ a 1D convolutional VQVAE to learn such a latent representation.

**Model.** As depicted in Fig. 3, our VQVAE consists of an Encoder $E_{\text{conv}}$, a Decoder $D_{\text{conv}}$, and a quantization module $Q$ using a codebook $V$. The model is inspired by [31, 48, 60]. The Encoder and Decoder, composed of 1D convolution layers, use two stride-2 convolutions and two 2 upscaling layers each, setting the upscaling and downscal-

ing rate $l$ to 4. The input motion $X \in \mathbb{R}^{T \times d}$ is encoded by the encoder in $Z = E_{\text{conv}}(X) \in \mathbb{R}^{T_z \times d_V}$, which is then quantized in $\hat{Z} \in \mathbb{R}^{T_z \times d_V}$. Note that $l$ denotes the temporal down-scaling factor of the mapping, $T_z := \lfloor T/l \rfloor$ denotes the length of the downscaled motion in the latent space. Also, $d$ and $d_V$ denote the dimensions of the single-frame human pose representation and the quantized latent space, respectively. Finally, $\hat{Z}$ is reconstructed as $\hat{X} \in \mathbb{R}^{T \times d}$ by the decoder.

**Quantization and optimization.** Our quantization $Q$ aligns with a discrete codebook $V = \{v_1, ..., v_C\}$, where $C$ represents the number of codes in the codebook and $v_i \in \mathbb{R}^{d_V}$. Specifically, each element $z_i$ of the latent vector sequence $Z = E_{\text{conv}}(X) = \{z_1, ..., z_{T_z}\}$ is quantized into the closest codebook entry $v_{s_i}$ with the corresponding codebook index $s_i \in \{1, ..., C\}$. Thus, our VQVAE can be represented by the following equation:

$$\hat{Z} = Q(Z) := \left[\arg\min_{v_{s_i}} ||z_i - v_{s_i}||_2\right]_i \in \mathbb{R}^{T_z \times d_V} \quad (1)$$

$$\hat{X} = D_{\text{conv}}(\hat{Z}) = D_{\text{conv}}(Q(E_{\text{conv}}(X))). \quad (2)$$

Eq. (2) is non-differentiable, and we handle it by the straight-through gradient estimator. During the backward pass, it approximates the quantization step as an identity function, copying gradients from the decoder to the encoder [6]. This allows the training of the encoder, decoder,

and codebook through optimization by following loss:

$$\mathcal{L}_{\text{VQ}} = \mathcal{L}_{\text{recon}}(X, \hat{X}) + ||sg\left[E_{\text{conv}}(X)\right] - \hat{Z}||_2^2$$
$$+ \beta||sg\left[\hat{Z} - E(X)\right] - \hat{Z}||_2^2. \tag{3}$$

The term $\beta||sg\left[\hat{Z} - E_{\text{conv}}(X)\right] - \hat{Z}||_2^2$, is referred to as a commitment loss, has shown to be necessary to stable training [53]. The reconstruction loss $\mathcal{L}_{\text{recon}}$ consists of L1-loss on the parameter, reconstructed joint, and velocity.

**Product quantization.** To enhance the flexibility of the discrete representations learned by the encoder $E_{\text{conv}}$, we employ a product quantization. Each element $z_i$ within $Z = E_{\text{conv}}(X)$ is divided into $K$ chunks $(z_i^1, ..., z_i^K)$, with each chunk discretized separately using $K$ different codebooks. The size of the learned discrete latent space increases exponentially with $K$, resulting in a total of $C^{TK}$ combinations, where $C$ is the size of each codebook. Although the increase in $T$ and $K$ provides a positive gain in both reconstruction quality and diversity, it introduces a trade-off that makes mapping text to latent space more challenging. The utility of using product quantization is empirically validated in our experiments.

## 3.2. Mapping a text onto discrete latent space

**Motivation.** We propose a Transformer-based Text Encoder that predicts a sequence of indices in discrete latent space given an input text and desired motion length $T$. At train time, the target sequences are obtained using the trained VQVAE by encoding ground truth target motions. One difficulty is that the input text is of variable dimension, a-priori independent of the length of the corresponding motion. To address this, we embed the conditioning signals and use a first Transformer block to inject that information into a sequence of $T_z$ positional embeddings, as illustrated in Fig. 4. Note that $T$ and $T_z$ denote the desired length in motion space and downscaled length in motion latent space, respectively. This yields a sequence of $T_z$ vectors, which are all functions of the input text and length. A second Transformer block, this time causal, then uses this information to perform autoregressive next index prediction, ultimately obtaining the predicted index sequence.

**Model.** As depicted in Fig. 4, our approach involves two Transformers, $H_1$ and $H_2$. To form the input for $H_1$, we first encode the text through CLIP [43] and a linear layer into $e_{\text{text}} \in \mathbb{R}^{d_H}$, and embed the desired length $T$ through the embedding layer $I_{\text{len}}$ into $e_{\text{len}} \in \mathbb{R}^{d_H}$, respectively. Note that $d_H$ denotes the input dimension of the Transformer layers. We concatenate $e_{\text{text}}$ and $e_{\text{len}}$, along the time dimension, following with positional embedding vectors $\text{PE}_1 \in \mathbb{R}^{T_z \times d_H}$ representing the temporal dimension in motion latent space. This is used as input to $H_1$; we discard the first two outputs along the time dimension and obtain

the text-length embedding

$$\{e_{\text{text-len}}^i\}_{i=0}^{T_z} \in \mathbb{R}^{T_z \times d_H} = H_1(e_{\text{text}}, e_{\text{len}}, \text{PE}_1)[2 : T_z + 2]. \tag{4}$$

The second Transformer block is used for autoregressive next index prediction. Given the previous indices, $\{s_i\}_{i=0}^{t-1} = (s_0 := s_\phi, s_1, ..., s_{t-1})$, and $\{e_{\text{text-len}}^i\}_{i=0}^{t-1}$, we estimate the distribution $p(s_t | \{e_{\text{text-len}}^i\}_{i=0}^{t-1}, \{s_i\}_{i=0}^{t-1})$. Each index $\{s_i\}_{i=0}^{t-1}$ is embedded through the embedding layer $I_{\text{idx}}$ into $\{e_{\text{idx}}^i\}_{i=0}^{t-1}$, concatenated with $\{e_{\text{text-len}}^i\}_{i=0}^{t-1}$. The concatenated input is added with positional embedding $\text{PE}_2 \in \mathbb{R}^{t \times 2d_H}$ and passed to the Transformer layer $H_2$. The output corresponding to $e_{\text{idx}}^{t-1}$ is then processed through a linear layer to estimate the likelihood,

$$p(s_t | \{e_{\text{text-len}}^i\}_{i=0}^{t-1}, \{e_{\text{idx}}^i\}_{i=0}^{t-1}). \tag{5}$$

During training, we utilize a causal mask, following PoseGPT [31], to handle this process in a single forward pass. At test time, we repeat the autoregressive sampling $T_z$ times to obtain the final indices $\{s_i\}_{i=1}^{T_z}$.

**Optimization goal.** This part of the model is trained to estimate the likelihood conditioned on the text and length input by minimizing the negative log-likelihood of the target indices under the output distribution.

## 3.3. Generation of long-term motion with T2LM

Fig. 2 gives an overview of how **T2LM** works at test time. Note that we use different notation in Sec. 3.3 from Secs. 3.1 and 3.2. Given a stream of sequential inputs $\{(w_i, T_i)\}_{i=1}^L$ of arbitrary length $L$, with $w_i$ and $T_i$ corresponding to the $i$-th ($i \in \{1, ..., L\}$) textual action description and desired motion length, respectively. We generate a corresponding realistic and smooth long-term motion, represented as a sequence of poses, $X_{\text{long}} \in \mathbb{R}^{(\sum_{i=1}^L T_i) \times d}$. Each pair of element $(w_i, T_i)$ is first individually passed to the Transformer Text Encoder to obtain a sequence $\{s_1^i, ..., s_{T_i/l}^i\}$ of discrete indices, where $l$ denotes the temporal down-scaling factor of the mapping. Then, the extracted discrete indices $\{\{s_j^i\}_{j=1}^{T_i/l}\}_{i=1}^L$ are dereferenced using the codebook $V$ and concatenated into a continuous sequence of latent vectors. This gives us the final input to the decoder:

$$\boldsymbol{Z} = \{V(s_1^1), \dots V(s_1^{T_1/l}) \dots, V(s_L^1) \dots V(S_L^{T_L/l}).\} \tag{6}$$

Finally, using a 1D convolutional decoder $D_{\text{conv}}$, we decode these latent vectors to obtain the desired long-term motion:

$$X_{\text{long}} = D_{\text{conv}}(\boldsymbol{Z}) \in \mathbb{R}^{(\sum_{i=1}^L T_i) \times d}. \tag{7}$$

Notably, the input of the convolutional decoder is a continuous stream of arbitrary length rather than independently generated actions that are later blended together.

| Trans. Vectors | $FID_{VQ}\downarrow$ | R-Prec.↑ | FID↓ | Diversity↑ | TS-FID↓ |
|---|---|---|---|---|---|
| 6 | 0.231 | 0.446 | 0.716 | 9.924 | 2.121 |
| 4 | 0.196 | **0.453** | 0.634 | 9.562 | 1.842 |
| 2 | 0.204 | 0.451 | 0.689 | 9.972 | 1.554 |
| **0 (Ours)** | **0.161** | 0.445 | **0.457** | **10.047** | **1.389** |

Table 2. **Ablation study on transition latent vectors.** We ablate the performance with respect to the size of transition latent vectors.

| Codebook Conf. | $FID_{VQ}\downarrow$ | R-Prec.↑ | FID↓ | Diversity↑ | TS-FID↓ |
|---|---|---|---|---|---|
| size 64 | 0.181 | **0.460** | 0.568 | 9.471 | 1.516 |
| size 128 | 0.156 | 0.389 | 1.751 | 9.33 | 1.670 |
| dim 128 | 0.418 | 0.417 | 0.761 | 9.600 | 1.822 |
| dim 256 | 0.246 | 0.447 | 0.767 | 9.707 | 1.620 |
| num 1 | 0.538 | 0.449 | 0.581 | 9.728 | 1.832 |
| num 4 | **0.062** | 0.424 | 0.515 | 9.289 | **1.325** |
| **256, 512, 2 (Ours)** | 0.161 | 0.445 | **0.457** | **10.537** | 1.389 |

Table 3. **Ablation study on codebook.** We ablate the performance with respect to the codebook configuration.

## 4. Experiment

### 4.1. Implementation details

For VQVAE, we used a codebook of 512 dimensions, $C = 256$ vectors in each $K = 2$ book for product quantization. We implement our framework with PyTorch [34]. Our Text Encoder is a Transformer with three layers, 2048 inner dimensions, and 16 multi-head attentions. We use AdamW [30] as an optimizer with a learning rate of 2e-4 and 3e-4, respectively, for training the VQVAE and Text Encoder. VQVAE and Text Encoder are trained for 1000 and 700 epochs, respectively, with the StepLR learning rate scheduler of step size 350 and a decrease rate of 0.5. The size of the mini-batch is set to 128. We applied a linear interpolation augmentation during VQVAE training and random corruption [60] augmentation for the Text Encoder. Training our model takes about a day on a single Nvidia 2080Ti GPU.

### 4.2. Dataset

We conducted experiments on two datasets: HumanML3D [15] and BABEL [41]. Our experiments focused mainly on the HumanML3D dataset to show the performance of our proposed T2LM without sequential training datasets, emphasizing its effectiveness in long-term generation. Regarding the BABEL dataset, we also compared our approach with existing long-term generation methods that rely on sequential data. Both datasets were evaluated using widely used evaluation protocols [15].

**HumanML3D.** The HumanML3D dataset comprises

| Category | Method | Sliding-scope | | Transition-scope | |
|---|---|---|---|---|---|
| | | FID↓ | Div.↑ | FID↓ | Div.↑ |
| - | **GT Motion** | 0.003 | 9.08 | - | - |
| Long-term (w.o. seq. data) | DoubleTake [46] | 1.23 | 7.824 | 1.753 | 7.499 |
| | **T2LM(Ours)** | **0.440** | **8.667** | **1.389** | **8.690** |

Table 4. **Comparison to SOTA: Long-term motion on HumanML3D test set.** We compare the long-term generation performance with the state-of-the-art method DoubleTake.

| Category | Method | Sliding-scope | | Transition-scope | |
|---|---|---|---|---|---|
| | | FID↓ | Div.↑ | FID↓ | Div.↑ |
| - | **GT Motion** | 0.005 | 9.53 | 0.078 | 8.53 |
| Long-term (with seq. data) | TEACH [5] | 2.633 | **9.236** | **2.173** | **9.429** |
| | MultiAct [22] | 3.128 | 8.593 | 3.694 | 8.338 |
| Long-term (w.o. seq. data) | DoubleTake [46] | 2.013 | 6.920 | 3.874 | 7.342 |
| | **T2LM(Ours)** | **1.799** | 9.06 | 3.535 | 7.941 |

Table 5. **Comparison to SOTA: Long-term motion on BABEL test set.** We compare the long-term generation performance with previous state-of-the-art methods.

14,616 motions, each associated with 3-4 textual descriptions. These motions, sampled at 20 FPS, originated from the AMASS and HumanAct12 motion datasets, with manual additions of text descriptions. During training, we used motions with lengths ranging from a minimum of 40 frames to a maximum of 196 frames.

**BABEL** We utilized the text version of the BABEL dataset [5]. This dataset includes 10,881 sequential motions, each annotated with textual labels for action segments. We used motions processed similarly to TEACH [5], with lengths ranging from a minimum of 44 frames to a maximum of 250 frames.

### 4.3. Evaluation metrics

**Sliding-scope and Transition-scope.** Existing evaluation metrics for motion generation rely heavily on extracting features from the entire motion, making them dependent on motion length and inadequate for quantitatively assessing the quality of generated long-term motions. We propose two new evaluation criteria to address this limitation: FID and diversity within a Sliding-scope and Transition-scope.

We use a fixed window of 80 frames for both scopes to extract subsets of long-term motions. We then measure FID and Diversity by comparing these subsets with sets extracted identically from the ground truth motion set. In Sliding-scope (SS-FID and SS-Div), we slide the window with a stride of 40 frames from the beginning to the end of the generated long-term motion to extract samples. In the Transition-scope (TS-FID and TS-Div), we extract samples centered around transitions in the generated long-term motion. The Sliding-scope provides an overall measure of how realistically the generated long-term motion represents

| Category | Method | R-Precision↑ | | | FID↓ | Diversity↑ | MM-Dist↓ |
|----------|--------|-------|-------|-------|------|-----------|----------|
| | | Top-1 | Top-2 | Top-3 | | | |
| - | **GT Motion** | 0.339 | 0.514 | 0.620 | 0.004 | 8.51 | 3.57 |
| Long-term (with seq. data) | TEACH [5] | - | - | 0.46 | 1.12 | 8.28 | 7.14 |
| | MultiAct [22] | 0.266 | 0.353 | 0.427 | 1.283 | 8.306 | 8.439 |
| Long-term (w.o. seq. data) | DoubleTake [46] | - | - | 0.43 | 1.04 | 8.14 | 7.39 |
| | **T2LM(Ours)** | **0.314** | **0.483** | **0.589** | **0.663** | **8.989** | **3.811** |

Table 6. **Comparison to SOTA: Single-action on BABEL test set.** We compare the generation performance of a single action to previous state-of-the-art methods.

| Category | Method | R-Precision↑ | | | FID↓ | Diversity↑ | MM-Dist↓ |
|----------|--------|-------|-------|-------|------|-----------|----------|
| | | Top-1 | Top-2 | Top-3 | | | |
| - | **GT Motion** | 0.511 | 0.703 | 0.797 | 0.002 | 9.503 | 2.974 |
| Long-term (w.o. seq. data) | DoubleTake [46] | - | - | 0.59 | 0.60 | 9.50 | 5.61 |
| | **T2LM(Ours)** | **0.445** | **0.631** | **0.731** | **0.457** | **10.047** | **3.311** |

Table 7. **Comparison to SOTA: Single-action on HumanML3D test set.** We compare the generation performance of a single action to previous state-of-the-art methods. Note that our main comparison target are only the long-term generation methods.

the entire sequence. At the same time, the Transition-scope evaluates how smoothly and seamlessly the long-term motion portrays transitions between actions. We use the pre-trained feature extractor from [15] to encode the representation of motion and text. We evaluate the quality of generated short-term action with R-precision, FID, MultiModal distance, and Diversity. Furthermore, we propose SS-FID and TS-FID to assess the quality of generated long-term motion quantitatively. **R-Precision.** For each motion, we rank the Euclidean distance to 32 text descriptions of 1 positive and 31 negatives. We report the Top-1, Top-2, and Top-3 accuracy. **FID.** We report the Frechet Inception Distance between the set of ground truth motions and generated motions. **MM-Distance.** We report the average Euclidean distances between the features of each text and motion. **Diversity.** We report the average Euclidean distances of the pairs in a set of 300 generated motions.

### 4.4. Ablation study

This section presents an ablation study on an alternative design idea using a transition latent vector and alternative configurations of the codebook in VQVAE. Quantitatively, it is conducted using five metrics: $FID_{VQ}$, R-Prec., FID, Diversity, and TS-FID. Note that $FID_{VQ}$ represents the FID score of the reconstructed motion by the VQVAE. Please refer to the supplementary material for other ablation studies.

**Transition latent vector.** We considered two ways of chaining a stream of latents from different texts at inference time. The first consists of simply concatenating the features; the second uses an additional token in the VQ-VAE codebook to denote transitions. For this second option, we add the learnable transition vectors in between latents of each text: $V(s^i_{\lfloor T_i/l \rfloor})$ and $V(s^{i+1}_1)$ at inference time as de-

picted in Fig. 2 and Sec. 3.3. To train these transition latent vectors, we randomly substitute part of the quantized latent vectors $\hat{Z}$ into the transition latent vectors while training the VQVAE. While using a transition latent is a very reasonable idea used in methods such as MultiAct [22] and Double-Take [46], empirically, we found that a technique based on concatenation works best while being more straightforward.

Tab. 2 presents the results. The leftmost column indicates the size of transition vectors; the length of the additional transition is $2 \times l$ if we use two transition vectors, where $l$ denotes the scaling rate of the VQVAE. Interestingly, the most straightforward approach of using concatenation (*i.e.*, first idea) performs best in our case. Specifically, a decrease in performance was observed as the size of transition latents increased in four metrics. The decrease in FID and Diversity, reflecting single-action quality, signals a reduction in the representation power of the latent space during transition latent training. This is evidenced by the decrease in reconstruction metrics for the VQVAE measured by $FID_{VQ}$. We conclude that using additional latents to represent transitions is not beneficial when sequential datasets are not employed, as evidenced by the degradation of TS-FID, which indicates transition quality.

**Codebook configuration.** In Tab. 3, we present quantitative measures for various codebook configurations used in the VQVAE. Commonly, an increase in the complexity of the codebook results in better performance of VQVAE reconstruction. However, this comes at the expense of more complicated predictions for the latent sequence prediction model. Indeed, it does not lead to monotonously improving final generations, which is clearly visible when using four codebooks. Given these results, we chose the setting with 2 codebooks, 256 vectors each, and 512 dimensions.

(a) "Wave hand" → "Walks in a circle" → "Runs forward"

(b) "Walks forward fast" → "Walks back" → "Putting a golf ball"

(c) "Walks backward, then walk forward to original position" → "Raise both arms and squat" → "Walks forward a couple steps, then turn back, walk back to the original position"

Figure 5. **Qualitative result**. We provide visualizations of generated long-term motions obtained with our method. The first, second, and third actions are rendered in blue, purple, and brown, respectively. *This is a video figure that is best viewed by Adobe Reader.*

## 4.5. Comparison to state-of-the-art

In this section, we compare the quality of motions generated with our **T2LM** to previous methods on the HumanML3D [15] and BABEL [41] datasets. Regarding the experiment on BABEL, we trained our model with individual actions and text annotations without using transitions. Our main comparison target on BABEL and HumanML3D is DoubleTake [46], the only long-term generation method trained without sequential data. Furthermore, we also compare with TEACH [5] and MultiAct [22] on BABEL dataset. [1] Our straightforward approach outperforms previous long-term generation methods in both single-action and long-term generation despite not requiring any sequential data for training.

**Long-term generation.** Tabs. 4 and 5 shows that our **T2LM** outperforms the main competing method, DoubleTake [46], in every criteria on both HumanML3D [15] and BABEL [41]. Regarding the *Sliding-scope* evaluation, our model demonstrates better overall quality of generated long-term motion compared to DoubleTake. Additionally, in the *Transition-scope* evaluation, our model produces more realistic transitions than those generated by DoubleTake. When evaluating long-term generation on the BABEL dataset, our model outperforms MultiAct on SS-FID, SS-Div. and TS-FID metric. Our method also shows the better performance compared to TEACH on the SS-FID metric, indicating better overall quality. However, ours showed inferior performance in the *Transition-scope* evaluation. This can be attributed to the usage of transitions

---

[1]ST2M is excluded from the comparison, since they do not use the 135-dimension representation as TEACH, DoubleTake and Ours. Instead, ST2M used 263-dimension representation. As a result, their quantitative evaluation lies on different dimension from TEACH, DoubleTake and Ours. (Quantitative scores of GT motions in [25] and [46] are different.)

from BABEL in TEACH during training time, while we train with individual actions only.

**Single-action generation.** Tabs. 6 and 7 show that **T2LM** outperforms previous long-term generation methods by a large margin on both HumanML3D [15] and BABEL [41]. Specifically, our T2LM scored 14.1% higher Top-3 R-precision compared to DoubleTake [46] on HumanML3D. Moreover, we gained 16.2%, 15.9% and 12.9% Top-3 R-precision over MultiAct [22], DoubleTake [46] and TEACH [5], respectively, on BABEL. Our superior performance is credited to the localized representative regions of each latent vector, combined with our Text Encoder, effectively conveying semantics from the text to the appropriate temporal dimensions.

## 4.6. Qualitative result

We present our generated long-term motion videos in Fig. 5. The video figure is best viewed by Adobe Reader. We downsampled the original video rendered in 24FPS into 6FPS and then displayed it in 15FPS. Please refer to the supplementary material for better visualization.

## 5. Conclusion

In this work, we proposed a conceptually simple yet effective long-term human motion generation framework by composing VQVAE and Transformer-based Text Encoder. Our approach achieved state-of-the-art performance compared to previous long-term generation methods on both actions and transitions. We also performed a detailed analysis on various model designs.

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
