# OpenReview forum: "T2LM: Long-Term 3D Human Motion Generation from Multiple Sentences"
_thecvf.com/CVPR/2024/Workshop/HuMoGen — CVPR 2024 Workshop HuMoGen Submission_

### Official Review · Reviewer_N4ss · 2024-03-28
**Text to Long-term motion generation. The model utilizies discrete diffusion using VQVAE and a text encoder.**

**Rating:** 4
**Confidence:** 5

**Review:**

Positively, the paper employs a sensible approach to quantization and mapping of latent representations, resulting in commendable qualitative outcomes. In addition the level of writing is high.

On the downside, the paper seems to extend motion techniques rooted in VQ-VAE, such as T2M-GPT. Its enhancements feel like engineering rather than innovations. In addition there are some inaccuracies that should be fixed for the final version: In Table 1, DoubleTake supports continuous generation just like T2LM, Equation 3 may contain errors, and there are some typos here and there.

Overall, I am inclined to accept this paper as it meets the standards of a workshop.

---

### Official Review · Reviewer_Xmu1 · 2024-04-01
**T2LM**

**Rating:** 4
**Confidence:** 4

**Review:**

T2LM is a method for generating long motions, described by a sequence of texts, each describing the motion of a single segment. The method is based on the GPT framework which is popular for motion generation and previously presented in the papers T2M, T2M-GPT, MotionGPT, and others. T2LM straightforwardly works at inference time by inputting the relevant text when its segment is generated.

Comments
- T2LM is generating the motions continuously without the need to stitch motion segments.
- Comprehensive evaluation - comparing to recent work, both diffusion and VAE-based solutions.
- T2LM outperforms recent works in terms of FID
- The novelty of this work is minimal, yet the method seems to work surprisingly well and is very relevant for this workshop.


Questions and suggestions for the next revision
- Figure 4 caption: test -> text (?)
- Figure 5 includes a single frame instead of motion, please fix that.

---

### Meta-Review · Area_Chair_YGiW · 2024-04-04

**Recommendation:** Accept

**Metareview:**

The paper addresses the task of long-motion synthesis.

Pros:
* Well written
* Thoroughly tested
* High qualitative results

Cons:
* Limited novelty
* Textual inaccuracies

**Guidance to authors:** Kindly address any textual inaccuracies highlighted by the reviewers, and correct any other typos or inaccuracies present in the paper.

---

### Decision · Program_Chairs · 2024-04-06

**Decision:**

Accept

**Comment:**

The paper will be published as part of the official CVPR workshop proceedings upon submission of the camera-ready version.